# Properties and Fungal Communities of Different Soils for Growth of the Medicinal Asian Water Plantain, *Alisma orientale*, in Fujian, China

**DOI:** 10.3390/jof10030187

**Published:** 2024-02-29

**Authors:** Xiaomei Xu, Wenjin Lin, Nemat O. Keyhani, Sen Liu, Lisha Li, Yamin Zhang, Xuehua Lu, Qiuran Wei, Daozhi Wei, Shuaishuai Huang, Pengxi Cao, Lin Tian, Junzhi Qiu

**Affiliations:** 1State Key Laboratory of Ecological Pest Control for Fujian and Taiwan Crops, College of Life Sciences, Fujian Agriculture and Forestry University, Fuzhou 350002, China; xuxiaomei@fjms.ac.cn (X.X.); m17633615410@163.com (S.L.); weidz888@sohu.com (D.W.); 2Fujian Key Laboratory of Medical Analysis, Fujian Academy of Medical Sciences, Fuzhou 350013, China; lisa_faoms@foxmail.com (L.L.); zym1099@fjms.ac.cn (Y.Z.); lxh6675@fjms.ac.cn (X.L.); 3Department of Biological Sciences, University of Illinois, Chicago, IL 60607, USA; keyhani@uic.edu; 4School of Materials Science and Chemical Engineering, Ningbo University, Ningbo 315211, China; wqr1880898@163.com; 5School of Ecology and Environment, Tibet University, Lhasa 850000, China; alaxender1989@126.com (S.H.); coparth@foxmail.com (P.C.); 6Tibet Plateau Institute of Biology, Lhasa 850000, China; tl15528534175@126.com

**Keywords:** soil fungal community, chemical properties, *Alisma orientale*, PacBio sequencing, *Alismatis rhizome*, heavy metal, pesticide residues

## Abstract

The Asian water plantain, *Alisma orientale* (Sam.) Juzep, is a traditional Chinese medicinal plant. The dried tubers of the *Alisma orientale*, commonly referred to as *Alismatis rhizome* (AR), have long been used in traditional Chinese medicine to treat a variety of diseases. Soil properties and the soil microbial composition are known to affect the quality and bioactivity of plants. Here, we sought to identify variations in soil fungal communities and soil properties to determine which would be optimal for cultivation of *A. orietale*. Soil properties, heavy metal content, and pesticide residues were determined from soils derived from four different agricultural regions around Shaowu City, Fujian, China, that had previously been cultivated with various crops, namely, Shui Dao Tu (SDT, rice), Guo Shu Tu (GST, pecan), Cha Shu Tu (CST, tea trees), and Sang Shen Tu (SST, mulberry). As fungi can either positively or negatively impact plant growth, the fungal communities in the different soils were characterized using long-read PacBio sequencing. Finally, we examined the quality of *A. orientale* grown in the different soils. Our results show that fungal community diversity of the GST soil was the highest with saprotrophs the main functional modes in these and SDT soils. Our data show that GST and SDT soils were most suitable for *A. orientale* growth, with the quality of the AR tubers harvested from GST soil being the highest. These data provide a systematic approach at soil properties of agricultural lands in need of replacement and/or rotating crops. Based on our findings, GST was identified as the optimal soil for planting *A. orientale*, providing a new resource for local farmers.

## 1. Introduction

Medicinal plants are defined as those applicable for human health maintenance and augmentation, as well as disease treatment and/or prevention, whose origins are often part of traditional medical practices [1,2]. China possesses one of the most abundant repositories of medicinal plants in the world, with a long history of use and relevant texts within the Chinese pharmacopeia. With the emergence of more organized and modern medical and agricultural practices, medicinal plants have transitioned from wild to cultivated species [3]. However, the cultivation of medicinal plants requires monitored conditions, as the production of bioactive molecules can vary, and the presence of environmental pollutants and toxins can impact their effectiveness safety. Thus, climate and soil conditions are crucial factors influencing the growth, development, and bioactivity of medicinal plants [4].

High concentrations of heavy metals, pesticide residues, plastics, and other toxic chemicals, as well as high levels of pathogenic microbes are key problems affecting the quality and usability of Chinese medicinal plants. In recent years, heavy metal contamination and pesticide residues in agricultural soil have attracted increased attention [5,6]. A recent national soil contamination survey in China revealed that roughly 16% of the agricultural land samples analyzed exceed the environmental quality standard, with heavy metals being the main pollutants (82.4%), especially cadmium and lead [7]. Thus, it is imperative to assess contamination and potentially toxic chemical residues in soils used for cultivating Chinese medicinal plants.

Microbes in agricultural soils are particularly important due to their impacts on ecological functions, biological stability [8,9], soil quality, and the ability to sustain plant growth [10,11]. As but one example relevant to heavy metal pollution, Zhou et al. [12] reported that inoculation of representative isolates of plant growth-promoting rhizobacteria (PGPR), including *Bacillus* and *Flavisolibacter* spp., into the soil effectively ameliorated cadmium toxicity in tomato plants. Overall, the properties of soil [13,14], the location [15], the types of vegetation [16], and the farming system [17,18], all contribute to soil microbial composition.

Fungi account for an important part of the soil microbiota [19]. Soil fungi are pioneers among decomposers, recycling carbon, nitrogen, phosphorus, and other compounds required for environmental cycling. In addition, they are indicators of the robustness of soil ecosystems, contributing to the maintenance of soil composition and fertility [20]. The fungal community structure and the dominant fungal populations exert reciprocal effects with plants, where each relies on and promotes the growth of the other (beneficial fungi), or in the case of fungal pathogens, can devastate or eliminate certain plants from local environments. Plant root exudates tend to enrich and select certain specific fungi. In turn, beneficial fungi can promote the absorption of soil nutrients by plants, increasing plant stress resistance and reducing disease, whereas pathogenic fungi can perform the opposite [21]. The plants, soil, and soil fungi constitute an ecosystem in dynamic equilibrium, where they restrict and influence each other; thus, knowledge concerning the fungal communities in different soils can lead to rational planting strategies for specific plants.

Shaowu City is located in the northwest of Fujian Province, and contains an important national commodity grain base, and key forestry areas including one of four forest product centers of Fujian Province. It is also an important growth and distribution center of Chinese medicinal plants in Fujian Province. In 2020, the local cultivation area of Chinese medicinal plants reached >3300 hectares, with a total production of ~0.38 million tons, and an output value of over USD15 million. In the past three years, the “Chinese experience” for COVID-19 prevention and control strategies has proven that use of certain traditional Chinese medicines can provide effective and economical strategies for the prevention of epidemic disease with low toxicity and side effects [22,23].

*Alisma orientale* (Sam.) Juzep (“Zexie” in Chinese, common name: Asian water plantain) is a well-known medical plant in China [24]. The dried tubers of the *A. orientale*, commonly referred to as *Alismatis rhizome* (AR), have long been used in traditional Chinese medicine to treat a variety of diseases, including dysuria, diarrhea, edema, stranguria, diabetes, and hyperlipidemia [25]. *A. orientale* is mainly cultivated in the Jian’ou and Shaowu regions of Fujian Province. In recent years, the agricultural industry in Shaowu City, including a planting and harvesting of a variety of grains, fruits, and tea, has faced a significant decline due to issues such as the cultivar aging and degradation. To promote the inheritance and development of traditional Chinese medicine, three administrative villages of Shaowu City were selected for determining their suitability for *A. orientale* cultivation: Shui Dao Tu (SDT, rice) and Guo Shu Tu (GST, pecan) in Gaofeng Village, Cha Shu Tu (CST, tea trees) in Shangqiao Village, and Sang Shen Tu (SST, mulberry) in Weimin Village. The objectives of this study were to: (1) characterize chemical properties (e.g., soil pH and organic matter content), heavy metal contamination, and chemical pesticide residue content of the selected soil regions; (2) characterize the soil fungal communities; (3) identify correlations between soil properties and fungal communities; and (4) test each soil for cultivation of *A. orientale* to determine which type of soil was most suitable.

## 2. Materials and Methods

### 2.1. Site Description and Soil Sampling

The four test plots were located in Shaowu, Fujian, China (117°02′–117°52′ N, 26°55′–27°57′ E), each of which had been previously cultivated with different agricultural plants (i.e., tea, mulberry, pecans, and rice). The plots were located at altitudes between 151.53 m and 483.80 m. The region has a subtropical monsoon climate. It is warm and humid, without severe cold in winter and extreme heat in summer. The average annual temperature is 17.7 °C, and the daily average temperature is ≥10 °C. The extreme maximum temperature is 40.4 °C, and the extreme minimum temperature is −8.1 °C. Moreover, the average annual evaporation and precipitation are 1802 mm and 1347 mm, respectively. The average annual relative humidity is 80%.

Twelve samples of bulk soil (three samples from each plot) were collected in the planting areas of the four different vegetation types. The samples were defined as follows: (1) Cha Shu Tu (CST soil, previously cultivated with tea trees) in Shangqiao Village (117°32′ N, 27°28′ E), (2) Sang Shen Tu (SST soil, previously cultivated with mulberry) in Weimin Village (117°68′ N, 27°07′ E), (3) Shui Dao Tu (SDT soil, previously cultivated with rice), and (4) Guo Shu Tu (GST soil, previously cultivated with pecan). The latter two soils were from Gaofeng Village (117°55′ N, 27°26′ E) (Figure 1). In each plot, the quadrat with a size of 1 cm × 1 cm was established, and the geographic location of the plot was recorded with GPS. A soil drill (3 cm in diameter) was used to drill holes along the diagonal of the plot to collect the bulk soil. Each plot was drilled with 16 holes. The soil from these 16 points was mixed to form a soil sample, and each soil sample was approximately 500 g. Stones, plant roots, and other litter were removed from the samples, which then were sealed in sterile bags. The soil samples were placed in an ice box and brought to the laboratory as soon as possible. The soil samples were divided into two parts: one aliquot of soil was immediately stored at −80 °C after 20 min in liquid nitrogen for DNA analysis; the other portion was sieved through a 2 mm mesh, naturally air-dried, and then analyzed for soil properties.

### 2.2. Soil Chemical Properties Analysis

The soil/water ratio for measuring pH was 1: 2.5 (*w*/*v*). An automatic nitrogen analyzer was used to determine total nitrogen (TN). Total phosphorus (TP) was measured by a UV spectrophotometer. Total potassium (TK) was determined by a flame photometer as described by Hengl et al. [26].

### 2.3. Heavy Metal Analysis and Pesticide Residue Estimation

The heavy metal analysis included the main potentially toxic elements, namely, cadium (Cd), chromium (Cr), mercury (Hg), nickel (Ni), plumbum (Pb), arsenic (As), cuprum (Cu), and zinc (Zn). The samples were digested in a mixture of perchloric acid and nitric acid at a ratio of 1:3 (*v*/*v*). Flame atomic absorption spectrometry was used to analyze the content of heavy metals. Two pesticide residues, benzene hexachloride (BHC) and dichlorodiphenyltrichloroethane (DDT), were tested. Pesticide residue estimation was performed using the gas chromatographic (GC) method. All the above detection methods were based on the GB15618-2018 risk control standard for soil contamination of agricultural land [27].

### 2.4. PacBio Long-Read Sequencing of Soil Fungi

Total genomic DNA was extracted from soil samples with the TGuide S96 Magnetic Soil/Stool DNA Kit (Tiangen Biotech (Beijing) Co., Ltd. Beijing, China) as directed by the manufacturer’s instructions. DNA integrity was measured with 1% agarose gels, and the DNA concentration of the samples was measured with the Qubit dsDNA HS Assay Kit and Qubit 4.0 Fluorometer (Invitrogen, Thermo Fisher Scientific, Waltham, MA, USA). All DNA with high quality was stored at −20 °C for further sequencing analysis. The ITS1F: 5′-CTTGGTCATTTAGAGGAAGTAA-3′ and ITS4: 5′-TCCTCCGCTTATTGATATGC-3′ universal primer set was used to amplify the full-length ITS rRNA gene from the genomic DNA extracted from each soil sample. Both the forward and reverse ITS primers carried sample specific PacBio barcode sequences to facilitate multiplexed sequencing. We chose to use barcoded primers to reduce chimera formation rather than the alternative of adding primers in the second PCR. KOD One PCR Master Mix (TOYOBOLife Science, Osaka, Japan) was used to perform 32 cycles of PCR amplification. The PCR amplification was started with initial denaturation at 95 °C for 5 min, followed by 8 cycles of denaturation at 95 °C for 30 s, annealing at 55 °C for 30 s, and extension at 72 °C for 45 s, followed by another 24 cycles of denaturation at 95 °C for 30 s, annealing at 60 °C for 30 s, extension at 72 °C for 45 s, and a final step at 72 °C for 4 min. The total PCR amplicons were purified with Agencourt AMPure XP Beads (Beckman Coulter, Indianapolis, IN, USA) and quantified using the Qubit dsDNA HS Assay Kit and Qubit 4.0Fluorometer (Invitrogen, Thermo Fisher Scientific, OR Waltham, USA). After the individual quantification step, equal amounts of amplicons were pooled. SMRTbell libraries were prepared from the amplified DNA using the SMRTbell Express Template Prep Kit 2.0 according to the manufacturer’s instructions (Pacific Biosciences, Menlo Park, USA). Purified SMRTbell libraries from the pooled and barcoded samples were sequenced on a single PacBio Sequel II 8M cell using the Sequel II Sequencing kit 2.0. Sequencing was performed with the aid of the BMK Cloud (Biomarker Technology Co., Ltd., Beijing, China). The generated sequences were compared by GenBank Blast search.

### 2.5. Bioinformatic Analysis

After exporting the PacBio disembarkation data as a CCS file, the following steps were performed. First, CCS reads were identified based on barcode sequences by the software lima v1.7.0 (https://github.com/PacificBiosciences/barcoding/, accessed on 7 March 2022), generating Raw-CCS sequences. Second, primer sequences were identified and removed by the software cutadapt 1.9.1 (https://doi.org/10.14806/ej.17.1.200, accessed on 7 March 2022). Raw-CCS sequences were filtered based on length, which generated clean-CCS sequences. Third, chimeric sequences were identified and removed byUCHIME v4.2 (http://drive5.com/uchime, accessed on 7 March 2022), generating effective CCS sequences.

The reads were clustered to obtain Operational Taxonomic Units (OTUs) with 97.0% similarity using Usearch software v10.0 [28]. Alpha diversity indexes (Chao, Ace, Simpson, and Shannon) were evaluated by QIIME2 (https://qiime2.org/, accessed on 7 March 2022). Beta diversity analysis was processed by QIIME software v2020.6 to compare species diversity between different samples. In addition, the vegan package in RStudio was used to visualize the similarity and dissimilarity of fungal communities among the soils of different plots through Principal Coordinates Analysis (PCoA) and Non-Metric Multidimensional Scaling (NMDS).

Redundancy analysis (RDA) was conducted based on the R language platform to reveal the relationship between community structure and soil properties. Funguild (Fungi Functional Guild) [29] was used as an annotation tool for the analysis of the fungal communities and to predict the soil fungal functional composition and phenotype [30,31].

### 2.6. Field Planting and Experimental Design

The field trial was conducted using *A. orientale* seeds obtained from Jian’ou, Fujian, the primary plantation base of *A. orientale*. The seeds were cultivated and transplanted in the soil of different plots. Seedlings were raised in July 2021 and harvested in early January of the following year. The spacing between plants and rows was 31.6 cm × 31.6 cm, with one plant per hole. Upon the maturity of *A. orientale*, 30 plants were harvested from each experimental soil plot. The medicinal part of *A. orientale*, known as *Alismatis rhizoma* (AR),was harvested, rinsed with clean water, and blotted with filter paper. The wet weight of AR was weighed. Subsequently, the tubers were dried at 60 °C to achieve a constant weight. The diameter of the tubers was measured using a vernier caliper, and the dry weight of individual tubers was recorded.

### 2.7. HPLC-MS Analysis for the Triterpenoid Content

Samples and reference standards (Alisol B-23 acetate, alisol B, Alisol C-23 acetate, alisol G, alisol F, alisol F 24-acetate, alisol A 24-acetate, and alisol A) were prepared following the procedure outlined in the general principles of the Chinese Pharmacopoeia (2020 edition). UPLC-MS analysis was conducted using a Waters XEVO-TQS Performance LC system, employing a Waters BEH C18 column (100 mm × 2.1 mm i.d., 1.7 µm, Waters, Milford, MA, USA) for separation. The flow rate was 0.3 mL·min^−1^, with an injection volume of 2 µL and a column temperature of 40 °C. The mobile phase consisted of 0.1% formic acid in water (A) and acetonitrile (B), following the gradient conditions: 1–1.5 min, 30–55% B; 1.5–5.5 min, 55–75% B; 5.5–7.5 min, 75–90% B; 7.5–8.5 min, 95% B; 8.5–8.6 min, 90–30% B; 8.5–10 min, 30% B. Mass spectrometric analyses were performed using a Waters XEVO-TQS system with an electrospray ionization source in positive ion mode. The following MS conditions were applied: cone gas flow, 150 L·h^−1^; desolvation gas flow, 800 L·h^−1^; desolvation temperature, 500 °C; source temperature, 150 °C; cone voltage, 40 V; capillary voltage, 3.50 kV. Argon was used as the collision gas.

### 2.8. Statistical Analysis

SPSS21.0 software (Chicago, IL, USA) was used to carry out the statistical analysis. Student’s t-test was performed between two groups and one-way analysis of variance with Student–Newman–Kuels multiple comparison was used for experiments involving more than two groups. The significance level for all tests was set at *p* < 0.05. Data on the soil properties and the contents of heavy metals, pesticides, and the triterpenoids were expressed as the mean ± standard deviation.

## 3. Results

### 3.1. Soil Chemical Properties

In terms of altitude, the areas ranged from 151 to 483 m above sea level, with GST > SDT > CST > SST. The chemical properties of the soil samples including pH, total nitrogen (TN), total phosphorus (TP), and total potassium (TK) were determined (Table 1), identifying several differences in the soil chemical properties of the different soils. In terms of pH, the soils were acidic, ranging from 4.49 to 5.36, with SST > GST > SDT > CST. The total nitrogen (TN) content of soils ranged from 0.03 to0.15% and showed the following trend: SDT > GST > CST > SST. With respect to total phosphorus (TP), the range in the soils was from 0.29 to 0.71 g/kg soil, with CST > SDT ≈ GST > SST. There were no significant differences in total potassium content (TK) among the different plots, which ranged from 13.6 to 25.7 g/kg soil.

### 3.2. Heavy Metal Analysis and Pesticide Residue Estimation

The concentrations of eight heavy metals, including cadmium (Cd), chromium (Cr), mercury (Hg), nickel (Ni), lead (Pb), arsenic (As), copper (Cu), and zinc (Zn) were measured in the soils of studied plots are shown in Table 2. Levels did not exceed the acceptable values established for the contamination of agricultural soil [28]. In addition, the soils were examined for two organic pesticides, benzene hexachloride (BHC) and dichlorodiphenyltrichloroethane (DDT), neither of which was found in detectable levels (Table 2).

### 3.3. PacBio Long-Read Sequencing Data

In the soil of studied plots, a total of 155,939 raw sequences, out of which 155,186 effective sequences were generated. The rarefaction curve [32] demonstrated adequate coverage of sequencing data, and that additional data would not generate significantly more OTUs (Figure 2A). These analyses showed that the amount of sequencing data was large enough to represent most species in the sample, and that only a limited number of new species would be found with any additional data. For the GST and SDT samples, all fungal curves reached a plateau when the sequencing depth increased to 12,000, and similarly for the CST and SST samples as the sequencing depth increased to 6000. The dilution curves of the GST and SDT samples were steeper than those of CST and SST, indicating that the fungal community diversities of GST and SDT were higher than those of CST and SST. There were 228, 577, 529, and 395 OTUs from the CST, GST, SDT, and SST samples, respectively. A total of 108 unique OTUs (47.37%) in CST, 229 (43.29%) in SDT, 232 (58.73%) in SST, and 251 (43.50%) in GST were observed. Among all fungal OTUs, 40 were shared (Figure 2B).

### 3.4. Soil Fungal Community Diversity

The soil fungal community richness indices (Chao1 and ACE) were GST > SDT > SST > CST (Table 3). Alpha diversity analyses indicated similar trends in Chao1 and ACE indices for the GST and SDT samples, with diversity within these samples significantly higher than that found in SST and CST samples. Shannon and Simpson indices reflect species diversity, and of the four soils examined, the Shannon diversity index in the GST samples was found to be the highest, with the Simpson index in the soil of CST determined to be the lowest; however, there were no significant differences among the different plots.

Beta diversity analyses were also performed to compare species diversity between the different samples. Here, we used principal component (PCoA) to reveal differences between the soil fungal communities, with two dimensional PCoA1 and PCoA2 accounting for 33.60% of the total variation (Figure 3A). Non-metric multidimensional scaling analysis (NMDS) was also applied to reveal the similarity of fungal community composition between the soils of from the different plots (Figure 3B). Compared to the SST and CST samples, the separation distance of the GST community was closer to the SDT sample, indicating that their fungal community structures were more similar, potentially as both were from Gaofeng Village.

### 3.5. Soil Fungal Community Distribution

From the sequencing analyses, a total of 13 fungal phyla, 43 classes, 99 orders, 173 families, 306 genera, and 419 species were obtained. The top four dominant phyla in the soils derived from the CST, GST, SDT, and SST plots were identical, although their relative abundance were different: *Ascomycota* (87.14%, 31.34%, 38.24%, and 77.78%), *Basidiomycota* (5.48%, 16.56%, 34.58%, and 6.54%), *Mortierellomycota* (1.59%, 22.10%, 4.61%, and 9.62%), and *Rozellomycota* (4.39%, 16.63%, 8.15%, and 2.00%) (Figure 4A). The majority of fungal OTUs belonged to the phylum *Ascomycota*. Overall, *Ascomycota* and *Basidiomycota* were the predominant phyla in the four soils examined. However, the phyla *Cryptomycota* and *Zoopagomycota* were unique to SDT and GST, and the phylum *Olpidiomycota* was only revealed in the SST soil. Overall, the SDT and GST samples had higher abundance at the phylum level compared to SST and CST.

LEfSe analyses showed that some fungal genera were significantly different between the different soil samples (Figure 4B). These data indicated that: (i) *Talaromyces*, *Racocetra*, *Aspergillus*, unclassified *Xylariales*, *Pseudothielavia*, and *Purpureocillium* were abundant in the SST soil; (ii) *Ascosphaera*, *Arthrocladium*,unclassified *Aspergillaceae*, *Acanthocorticium*, and *Lecanicillium* were abundant in the CST soil; (iii) *Ustilaginoidea* and *Conioscypha* were abundant in the SDT soil; and (iv) *Oidiodendron*, *Bullera*, *Mycosphaerella*, *Discosia*, *Rhizophagus*, *Neurospora*, and *Ramularia* were abundant in the GST soil.

Analysis of co-occurrence networks revealed a strong correlation between the fungal genera and the different soil samples (Figure 5). The analyses indicated that 29 fungal genera (e.g., *Microascus*, *Madurella*, *Latorua*, *Olpidium*, and others) were in the same module and found in SST soil, and 18 fungal genera (including *Betamyces*, *Rhodotorula*, *Scutellinia*, and *Sistotrema*) were found in the same module and were present in SDT soil. Overall, GST and SDT soils had similar fungal genera compositions, aligning with the findings of the soil fungal community diversity and the abundance at the phylum level analyses.

### 3.6. Relationship between Abundant Fungal Genera and Diversity Indices, and Soil Chemical Parameters

Redundancy analysis (RDA) based on the Bray–Curtis distance and Pearson correlation analysis was conducted to examine correlations between soil chemical parameters and the top 10 abundant fungal genera. The first two axes of the RDA explained 19.29% and 13.21%, respectively, of the total fungal variation. In the SDT soil, fungal genera were mainly affected by soil nutrients and had the highest correlation with soil TN content, followed by altitude. Fungal genera in the SDT soil had the lowest correlation with soil TP content. Soil nutrients mainly affected the genera *Boothiomyces*, *Saitozyma*,and *Penicillium* in the CST and SST soil samples, with pH as a key factor, mainly affecting the *Ascosphaera*, *Talaromyces*, and *Aspergillus.* In GST soil, TK content was a key factor and mainly affected the genera *Cladosporium*, *Mortierella*, and *Humicola* (Figure 6A).

The relationships between fungal diversity indices and soil chemical parameters were also analyzed (Figure 6B). The correlation between soil chemical parameters and the fungal community α-diversity indices reached a significant level with TN content showing a significant positive relationship with Chao1 and ACE, and the fungal community α-diversity showing a positive correlation with soil pH and TK content. Conversely, soil TP content exhibited a negative relationship with the fungal community α-diversity indices. In addition, altitude and TN content showed a positive relationship with Chao1 and ACE but a negative correlation with Simpson indices.

### 3.7. Functional Prediction of Fungal Phenotype by FUNGuild

FUNGuild (Fungi Functional Guild) was used as an annotation tool to parse the identified fungal communities into ecological lifestyles, namely, pathotrophy, symbiotrophy, and saprotrophy, which could be further subdivided into 12 subclassifications termed guilds. The relative abundances of these different trophic modes in the CST, GST, SDT, and SST soils were as follows: saprotrophs (18.07%, 67.12%, 86.59, and 36.01%, respectively), pathotrophs (72.50%, 13.29%, 5.64%, and 54.90%, respectively), and symbiotrophs (9.43%, 19.58%, 8.08, and 9.08%, respectively) (Figure 7A). Pathotrophs were the most abundant trophic mode found in CST and SST soils, whereas saprotrophs were the most abundant trophic mode found in the GST and SDT soils.

More detailed functional prediction classifications were performed for annotated fungi with relative abundances >1% (Figure 7B). In the CST soil, animal pathogens were the dominant guild (~59.2%), followed by fungal parasites (10%). In the GST soil, wood saprotrophs were the dominant guild (20.7%), followed by ectomycorrhizae and plant pathogens (11.4% and 10.9%, respectively). In SDT soil, wood saprotrophs were the dominant guild (13.2%), followed by litter saprotrophs and soil saprotrophs (11.2% and 10.9%, respectively). In SST soil, animal pathogens were the most dominant guild (31.3%), followed by litter plant pathogens (18.2%).

### 3.8. The Morphological Characteristics and Quality of the Alismatis Rhizoma Planted in Soils of Different Plots

*Alismatis rhizome* (AR), derived from the tuber of *A. orientale*, is used in traditional Chinese medicine to treat a variety of diseases. In terms of morphology, AR displaying a large overall size (~2–7 × 2–6 cm), yellow-white color, solid texture, and high powder content, are of high quality [33]. AR predominantly contain a suite of putatively bioactive terpenes, specifically protostane-type triterpenes and guaiane-type sesquiterpenes [34]. The triterpenoid components, including alisol B23-acetate, alisol B, and alisol C23-acetate, serve as the primary pharmacologically active compounds in AR [35]. In this study, we observed that AR cultivated in GST displayed robust growth with the maximum diameter value and heaviest weight, followed by those planted in SDT and SST, while AR planted in CST exhibited the smallest diameter and lightest weight (Table 4). In addition, we detected the contents of eight major triterpenoids. The results showed that the contents of alisol B 23-acetate, alisol B, and alisol C 23-acetatewere highest in AR planted in GST, with total contents ranging from 0.715 to 2.131 mg/g, 0.121 to 0.268 mg/g, and 0.0315 to 0.086 mg/g in the different soils, for each compound, respectively. AlisolG, alisol F, alisol F 24-acetate, alisol A 24-acetate, and alisolA, were all detected in low amounts, and most were not significantly different in quantity between AR grown in the four different plots. However, the content of total triterpenoids was the highest in AR grown in GST, followed by SDT (Table 4).

## 4. Discussion

Cultivation of medicinal plants can provide important economic and human health resources, particularly for small- to medium-scale indigenous farmers. In some cases, areas where growth of current crops are no longer sustainable, transitioning to cultivating medicinal plants can provide important new opportunities on the local level. In this regard, several important soil factors should be considered, including (1) soil physiochemistry; (2) levels of basic nutrients, e.g., nitrogen, phosphorus, and potassium; (3) levels of pollutants, including heavy metals, pesticide, and/or other toxins; and (4) the soil microbiome that can impact plant health and stress resistance. As to the pollutants, it is well known that high concentrations of heavy metals and pesticides have an adverse impact on both soil and plant health [36], and that these toxins can enter the food chain through water–soil–crop ecosystems, posing a serious threat to human health [37,38,39]. Accumulation of heavy metal and other toxins in soil can deteriorate soil properties and hinder plant growth or render plants unfit for human consumption [40]. Our data show that soils from the four tested plots in and around Shaowu City, Fujian province, previously used to grow various other crops did not contain significant levels of any of the toxins (heavy metal and pesticides) examined, indicating that at least with respect to this issue, all of the soils met the human safety requirements for cultivating medicinal plants.

Fungal diversity is affected by changes in season, soil chemical properties, and vegetation types, and inversely, with the exception of season, affects these parameters [41]. For example, the studies on the Northern Tibet alpine meadow suggested that soil pH and plant biomass and diversity can regulate changes in soil fungal community along the elevation gradients [42], while another study in the Italian Alps showed that the changes in soil fungal community were mainly governed by soil pH and the ratio of carbon to nitrogen (C/N) [43]. The variation in ectomycorrhizal fungi along the elevation gradient was closely related to soil temperature and moisture on Cairn Gorm mountain [44]. Plant biomass, soil temperature and moisture, soil ammonium nitrogen and nitrate nitrogen, available phosphorus, etc., can affect, and even predominantly determine, the soil fungal community diversity. In our study, we also observed a strong correlation between fungal richness and nitrogen concentration. However, the soil pH did not play a significant role in shaping the fungal communities, since α-diversity indices had no significant relationship with soil pH. The reason may be that all soils from the four tested areas were all acidic, with pH values ranging from 4.49 to 5.36, and the difference in pH values between different soils was not significant. Thus, the role of pH in shaping soil fungal diversity was not conclusive; instead, it collaborates with other soil characteristics to influence soil fungal diversity. The available research indicated that pH gradient was less strongly correlated to fungal community composition than bacteria, as fungi species typically had a wide pH optimum, often covering 5–9 pH units without experiencing significant growth inhibition [45,46]. The results from previous studies were thus corroborated by our results.

Soil microbial diversity can affect soil nutrient metabolism and soil fertility [47]. Typically, increased soil microbial diversity reflects improved soil quality, implying better substrate utilization and nutrient supply (for plants). Fungi are important members of the soil microbiome, and have profound effects on plant growth, health, resilience, stress response, and even plant secondary metabolite production. Thus, soil fungal diversity and content can be used as an important indicator of soil quality and in evaluating soil ecosystem functions [48]. Our data show that of the four different regions around Shaowu City sampled, the fungal community distributions in GST and SDT soils (both from around Gaofeng Village) were similar, with high species richness and community diversity. Various soil chemical properties, e.g., pH, and total nitrogen, carbon, and phosphorus, of the two plots were also similar, and both exhibited high fertility with ample local water sources. GST soil was originally used for rice cultivation before pecan trees were subsequently planted, potentially accounting for the greater soil fungal diversity seen in this soil as compared to the others. Conversely, SST and CST samples were derived from less fertile soils with lower water sources, correlating with their observed lower fungal diversity. CST soil was a kind of red clay, characterized by compact structure, poor ventilation, unfavorable drainage, and slow nutrient decomposition. This kind of soil was found to be unsuitable for plant growth, leading to poor plant quality, low fruiting rates, and increased susceptibility to disease. The quality of *A. orientale* planted in the four soil plots revealed that CST soil was the least suitable for *A. orientale*, while GST soil (with the highest fungal diversity) allowed for robust *A. orientale* growth.

In terms of fungal diversity, the FUNGuild functional prediction results showed that saprotrophs and pathotrophs were the predominant trophic modes in all the soils examined. Saprotrophic fungi are primary decomposers, regulating nutrient cycling in soil [49]. Fungal saprotrophs also contribute to the protection of crops from fungal pathogens and facilitate soil health [50,51]. Conversely, pathotrophic fungi could potentially cause plant disease [52]. Pathotrophic fungi gain nutrients by invading host cells and can damage or kill target plants. For soil, optimized nutrient management practices can significantly increase the number of saprotrophic fungi while decreasing the relative abundance of pathotrophic fungi, resulting in improved crop growth [53]. However, important differences in the abundance of pathotrophic fungi were seen, with higher levels in CST and SST soils as compared to GST and SDT soils. Conversely, the abundance of saprotrophic fungi was higher in the GST and SDT soils compared to the former two soils. In addition, mycorrhizal fungi play vital roles in facilitating plant access to essential nutrients [54]. Specifically, arbuscular mycorrhizas (AMs) form a mutualistic relationship with host plants, enabling the plants to acquire phosphorus and mineral nutrients, while receiving carbohydrates and lipids from the plants in return [55]. Furthermore, mycorrhizal fungi can enhance the host’s resistance to both biotic and abiotic stresses [56]. Notably, our study revealed that the proportion of mycorrhizal fungi in the GST soil was significantly higher than what was found in the other three soils. Finally, direct cultivation experiments in the different soils demonstrated that the GST soil was provided for more healthy and sustainable cultivation of *A. orientale* as compared to the other soils examined.

In summary, this study provides a land screening design combining characterization of soil chemical properties and fungal diversity to identify suitable soils for cultivation of medicinal plants. These parameters were verified by cultivation of *A. orientale* in the various soils, which demonstrated a strong correlation between the ability to sustain growth of the plant and fungal diversity and lifestyle analyses. We demonstrate specific sites in Shaowu city based on variations in the soil fungal community combined with soil properties that would be optimal for maintaining growth of *A. orientale*. These data also provide a basis for future studies aimed towards exploring functional aspects of the fungal soil microbiome as it relates to cultivation of specific crops and the potential for manipulating fungal diversity to increase fertility of less fertile soils.

## Figures and Tables

**Figure 1 jof-10-00187-f001:**
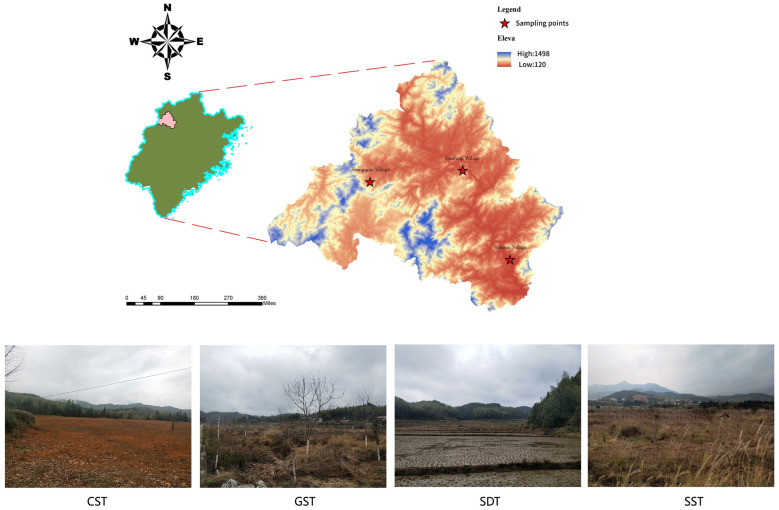
Location of the sampling sites and layout of the plots for the different soils. CST, SST, SDT, GST, the sites previously planted by tea trees, mulberries, rice, and pecan, respectively.

**Figure 2 jof-10-00187-f002:**
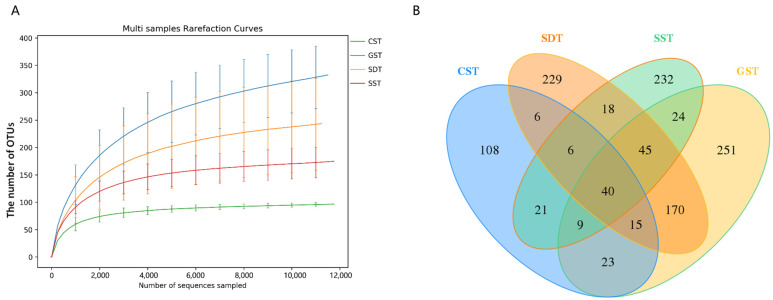
PacBio long-read sequencing data. (**A**) Rarefaction curves of soil samples from different plots. The *x*-axis represents the counts of randomly sampled sequences, and the *y*-axis represents the counts of OTUs detected by giving sequences. (**B**)Venn diagram of fungal communities in the soil of studied plots.

**Figure 3 jof-10-00187-f003:**
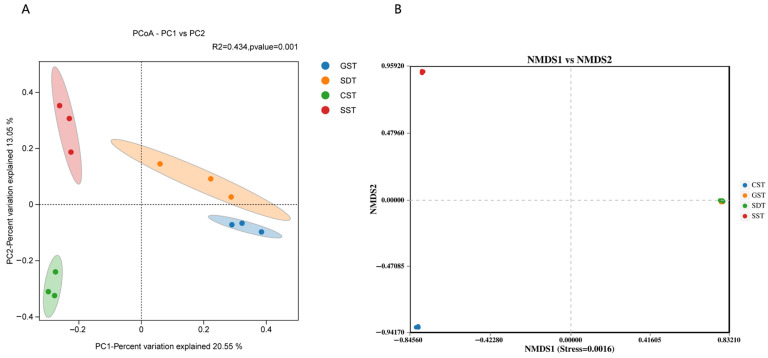
β-diversity analysis of four different soil samples. (**A**) The PCoA analysis for the difference in fungal community structure among the different samples. (**B**) Non-metric Multidimensional Scaling (NMDS) analysis for the similarity of fungal community structure among the different samples. Each dot represents a sample. Samples in different groups are presented in different colors. The *x*-axis and *y*-axis represent two eigenvalues that could maximize the difference between samples. The influence of each eigenvalue was measured as a percentage. *n* = 3.

**Figure 4 jof-10-00187-f004:**
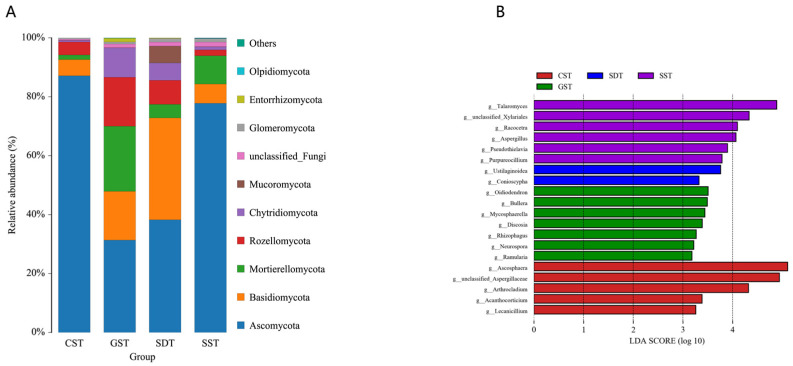
Analysis of the soil fungal community structure and composition in the CST, GST, SST, and SDT soil samples. (**A**) Community structure and composition at the phylum level. (**B**) Community structure and composition at the genus level. OTUs with an abundance of less than 1% were classified as ‘others’.

**Figure 5 jof-10-00187-f005:**
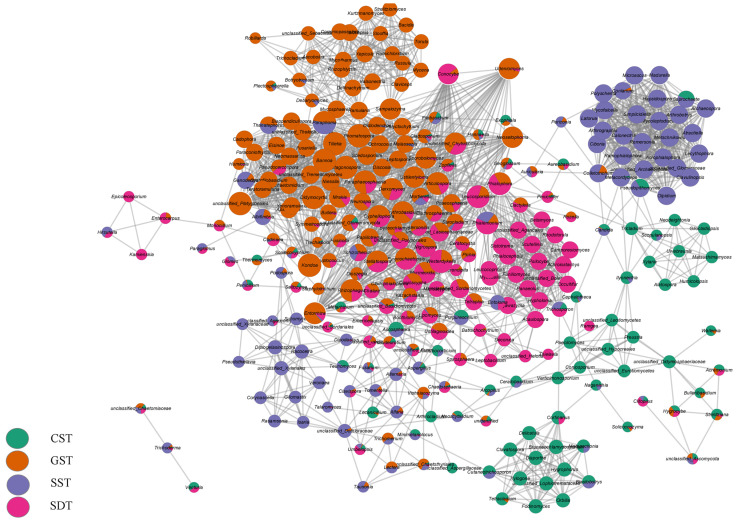
Correlation network analyses for the fungal genera in the soil of studied plots. Each node represents genus level assignments, and the edges represent correlations between genus pairs. The nodes are colored according to the different plots.

**Figure 6 jof-10-00187-f006:**
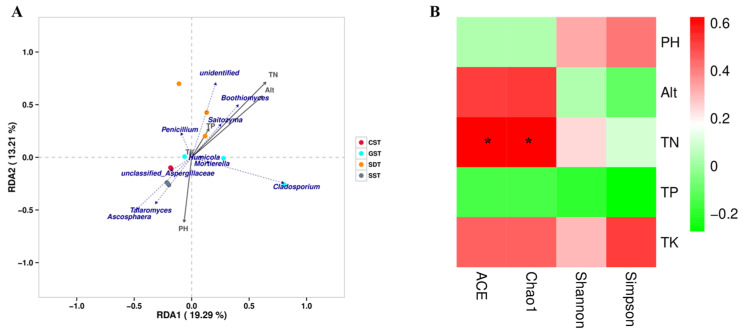
Relationship between abundant genera and diversity indices with soil chemical parameters. (**A**) Redundancy analysis (RDA) showing the relationship of abundant genera and diversity indices with soil chemical parameters among the soils of different plots. Obtuse angle represents a negative correlation, and acute angle between rays represents a positive correlation. (**B**) Correlation analyses between diversity indexes of fungal communities and soil chemical properties. ‘*’ indicates a significant correlation (*p* < 0.05). A positive and negative correlations are indicated in red and green, respectively.

**Figure 7 jof-10-00187-f007:**
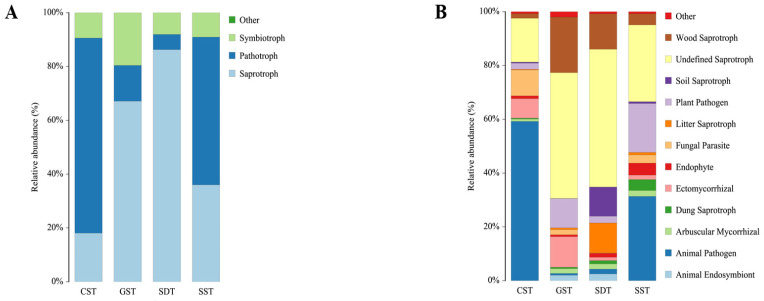
FUNGuild classification of soil fungal communities. (**A**) Trophic mode. (**B**) Guide detailed classification.

**Table 1 jof-10-00187-t001:** Soil chemical properties in the different study plots.

Taxon	CST	GST	SDT	SST
pH	4.49 ± 0.10 b	4.83 ± 0.10 ab	4.55 ± 0.10 ab	5.36 ± 0.11 a
Alt (m)	343.17 ± 1.02 ab	483.80 ± 2.0 a	464.10 ± 8.89 ab	151.53 ± 2.77 b
TN (%)	0.08 ± 0.001 ab	0.133 ± 0.02 ab	0.15 ± 0.03 a	0.03 ± 0.001 b
TP (g/kg)	0.71 ± 0.01 b	0.48 ± 0.10 ab	0.54 ± 0.13 ab	0.29 ± 0.01 a
TK (g/kg)	13.6 ± 0.03 a	25.7 ± 7.57 a	19.93 ± 4.56 a	24.3 ± 1.04 a

Note: The values are presented as the mean ± SD (*n* = 3), and different lowercase letters in the same line indicate significant differences among plots (*p* < 0.05).

**Table 2 jof-10-00187-t002:** The concentrations of heavy metals and organic pesticides in the soil of studied plots.

Group	Standard Values ^a^	CST	GST	SDT	SST
Cd (μg/g)	≤0.30	0.03 ± 0.002	0.09 ± 0.01	0.06 ± 0.01	0.14 ± 0.04
Cr (mg/kg)	≤250	70 ± 3.00	16 ± 2.52	25 ± 14.11	20 ± 2.01
Hg (mg/kg)	≤0.50	0.38 ± 0.10	0.20 ± 0.05	0.17 ± 0.03	0.03 ± 0.004
Ni (μg/g)	≤60	35.90 ± 1.00	8.50 ± 1.53	13.2 ± 6.93	8.10 ± 0.02
Pb (μg/g)	≤80	19.70 ± 0.30	24.20 ± 6.01	40.60 ± 4.98	35.8 ± 2.30
As (mg/kg)	≤30	11.9 ± 0.70	2.92 ± 0.41	1.93 ± 0.50	1.49 ± 0.30
Cu (μg/g)	≤50	29.6 ± 0.40	14.50 ± 0.91	13.00 ± 3.58	9.20 ± 0.51
Zn (μg/g)	≤200	117 ± 2.00	41.10 ± 5.45	46.70 ± 4.47	71.6 ± 1.03
DDT (mg/kg)	≤0.10	ND (<0.01)	ND (<0.01)	ND (<0.01)	ND (<0.01)
BHC (mg/kg)	≤0.10	ND (<0.01)	ND (<0.01)	ND (<0.01)	ND (<0.01)

Note: The values are presented as the mean ± SD (*n* = 3). ND stands for no detection. ^a^ The standard values are from the soil environmental quality standard of the People’s Republic of China (GB15618-2018) [28].

**Table 3 jof-10-00187-t003:** α-Diversity of fungal communities in indicated soil samples.

Group	ACE	Chao1	Simpson	Shannon
CST	99.65 ± 5.98 a	100.89 ± 8.17 a	0.75 ± 0.10 a	3.34 ± 0.87 a
GST	374.12 ± 80.36 b	377.14 ± 79.83 b	0.90 ± 0.07 a	5.02 ± 1.47 a
SDT	263.60 ± 140.90 abc	264.08 ± 141.7 abc	0.83 ± 0.16 a	4.33 ± 1.69 a
SST	182.05 ± 44.07 c	189.88 ± 31.45 c	0.91 ± 0.05 a	4.73 ± 0.69 a

Note: The values are presented as the mean ± SD (*n* = 3), and different lowercase letters in the same column indicate significant differences between groups (*p* < 0.05).

**Table 4 jof-10-00187-t004:** The morphological characteristics and contents of the eight triterpenoids in AR collected in different studied plots.

Group	CST	GST	SDT	SST
Diameter (mm)	36.87 ± 2.67 b	44.97 ± 4.21 a	42.77 ± 1.62 a	38.80 ± 1.41 ab
Wet weight (mg)	37.29 ± 0.71 c	81.2 ± 11.41 a	63.42 ± 4.76 b	55.54 ± 6.22 b
Dry weight (mg)	10.24 ± 1.59 b	25.20 ± 7.44 a	22.32 ± 0.72 a	14.69 ± 5.50 ab
AlisolB23-acetate (mg/g)	0.7150 ± 0.0168 b	2.1314 ± 0.5424 a	1.5233 ± 0.1264 c	0.8595 ± 0.0266 b
Alisol B (mg/g)	0.1211 ± 0.006 b	0.2680 ± 0.030 a	0.1613 ± 0.011 c	0.1470 ± 0.011 bc
AlisolC23-acetate (mg/g)	0.0315 ± 0.0016 b	0.0862 ± 0.0065 a	0.0721 ± 0.0034 c	0.0444 ± 0.0011 d
AlisolG (mg/g)	0.000	0.0072 ± 0.001	0.0030 ± 0.0003	0.0021 ± 0.0004
Alisol F (mg/g)	0.000	0.0005	0.0001	0.000
AlisolF 24-acetate (mg/g)	0.000	0.0003	0.000	0.000
AlisolA24-acetate (mg/g)	0.0002 ± 0.0001	0.0025 ± 0.0006	0.0005 ± 0.0003	0.0001 ± 0.0002
AlisolA (mg/g)	0.0003 ± 0.0001	0.0031 ± 0.0009	0.0014 ± 0.0006	0.0015 ± 0.0008
Total triterpenoids (mg/g)	0.8685 ± 0.0145 c	2.4991 ± 0.5450 a	1.7613 ± 0.1190 b	1.0547 ± 0.0267 c

Note: The values are presented as the mean± SD (*n* = 3), and different lowercase letters in the same line indicate significant differences between groups (*p* < 0.05).

## Data Availability

The datasets generated during the current study were deposited and are available at the National Center for Biotechnology Information (NCBI) Sequence Read Archive (SRA) under accession number PRJNA874184 (www.ncbi.nlm.nih.gov/sra/PRJNA874184, accessed on 27 August 2022). Other data generated or analyzed during this study were included in this published article and its additional files.

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
