# Peer review of "Properties and Fungal Communities of Different Soils for Growth of the Medicinal Asian Water Plantain, Alisma orientale, in Fujian, China"

_jof, 2024, doi:10.3390/jof10030187_

Round 1

Reviewer 1 Report

Comments and Suggestions for Authors

Summary

Line 19-21 confusing sentence.

The objective and conclusion are not well defined in the summary

Introduction

Update information line 59 is from 2014?

The introduction of line 37 to 101 presents unnecessary information and could be summarized.

Material and methods

Figure 1 the caption cannot be read.

In item 2.4, it was missing how the sequences were compared, which database was used.

Results

3.1 Line 255-258 is methodology.

A lot of methodology information is in the results and it is recommended to be removed.

Discussion

Line 471-474 is not a discussion already placed above in the text.

The discussion has citations but little discussion of the data obtained. What groups of fungi are present? Soil conditions for growing A. orientale? The conclusion of the work isn´t clear.

Author Response

Dear Editors and Reviewers:

Thank you for your letter and comments relating to our manuscript entitled “Physicochemical Properties and Fungal Communities of Soil Types for Growth of the Medicinal Asian Water Plantain, Alisma orientale” (ID: jof- 2855766). The comments were very helpful in revising and improving our manuscript as well as emphasizing the significance to our research. We have read the comments carefully and made corrections accordingly. Revised portions are marked in blue in the manuscript. The main corrections in the paper and our responses to the reviewer’s comments are given below. We hope that the revisions in the manuscript and our accompanying responses will be sufficient to make our manuscript suitable for publication in the Journal of Fungi.

Responses to the comments of the reviewer:

Reviewer 1#

Comments 1: Line 19-21 confusing sentence.

Response 1: We have changed the sentence expression in the revision.

Comments 2: The objective and conclusion are not well defined in the summary

Response 2: Thank you very much for your proposal. We have changed the corresponding part in the revised manuscript.

Introduction

Comments 3: Update information line 59 is from 2014?

Response 3: We have updated the information and citation in the revised manuscript.

Comments 4: The introduction of line 37 to 101 presents unnecessary information and could be summarized.

Response 4: We have revised the relative part of the introduction.

Material and methods

Comments 5: Figure 1 the caption cannot be read.

Response 5: We have changed Figure 1 in the revised manuscript.

Comments 6: In item 2.4, it was missing how the sequences were compared, which database was used.

Response 6: We have added “The generated sequences were compared by GenBank Blast search” in item 2.4.

Results

Comments 7: 3.1 Line 255-258 is methodology.

Response 7: We have deleted the sentences of Line 255-258 from 3.1 in the revised manuscript.

Comments 8: A lot of methodology information is in the results and it is recommended to be removed.

Response 8: We have removed the methodology information from the results in the revised manuscript.

Discussion

Comments 9: Line 471-474 is not a discussion already placed above in the text.

Response 9: We have revised the relative part of the discussion in the revised manuscript.

Comments 10: The discussion has citations but little discussion of the data obtained.

Response 10: We have improved the discussion in the revision.

Comments 11: What groups of fungi are present? Soil conditions for growing A. orientale? The conclusion of the work isn´t clear.

Response 11: We have revised the Discussion in the amended manuscript.

We tried our best to improve the manuscript and made some changes marked in blue in revised paper which will not influence the content and framework of the paper. We appreciate for Editors/Reviewers’ warm work earnestly and hope the revision will meet with your approval. Once again, thank you very much for your comments and suggestions.

Kind regards,

Junzhi Qiu

E-mail address: [email protected]

Reviewer 2 Report

Conducting this study, the authors did a lot of work on soil quality, fungal communities' composition, and the quality of medicinal plant Alisma orientale grown in the soil of four experimental plots. However, the significance of this work is greatly diminished by the low quality of results' presentation and their interpretation.

One of the major problem of the paper is its language, even not English by its own (although it needs to be thoroughly checked and corrected), but the scientific language. The way, in which the results of statistical analyses and the composition of fungal communities (all these "total fungal taxa") were described, arises great doubt about mycological background of the authors.  

The problems start with the title. The title of the paper submitting to the mycological journal begins with "physicochemical properties", which is illogical. Then, here and further in the text – the authors did not measure any soil physical parameters (color, texture, structure, porosity, density, consistence, aggregate stability, and temperature), only chemical parameters. Also, here and further in the text – the term "soil type" is from the soil classification and is not appropriate in the context of the study. The title also should denote the region of the study. Overall, the expression "Physicochemical Properties and Fungal Communities of Soil Types for Growth …" is awkward.

Abstract is also problematic because of the following reason. Abstract should be understood by itself, without the full paper reading. However, this is not the case of the submitted abstract. The mention of soil fungal communities (line 21) arises suddenly, without any association with the previous text and the optimal condition for the plant growth. It is unclear what all these "soil groups" mean (lines 23-24). It is also unclear what Alismatis rhizome means.

Introduction should not contain all these unnecessary repetitions and general statements (see the attached PDF file).

Subsection 2.1 - natural vegetation and soil type (types) should be denoted.

Subsection 2.8. – statistical analysis is unclearly written and should be rewritten.

Subsection 3.1. - There is no need to repeat the written in M&M! It is more than enough to write that the chemical properties of studied soils are presented in Table 1.

Subsection 3.3. - All these comparisons including "total fungal taxa" look meaningless, because "total taxa" is something borderless and vague.

Figure 2. – The combination of four different figures in one makes each figure hardly readable. Figure A looks unclear and meaningless. Figure C just presents number of OTUs in the soil of studied plots (what does "feature number" mean?). Figure D is meaningful only if the numbers in the diagram ovals present numbers of OTUs.

Discussion suffers from the superficial interpretation of the results obtained, logical gaps, unnecessary repetitions, and problematic statements (see the attached PDF file).  It is unclear for which purposes the authors studied the composition of fungal communities at different taxonomic levels, if they did not discuss this aspect at all, except for the extremely general, problematic, and unclear statements on lines 485-489. It looks like that in order to choose the best plot for growing Alisma orientale, it was quite enough to conduct the experiment described in the subsection 3.8.

All other numerous comments, corrections, and suggestions are inserted into the attached PDF version of manuscript. 

Author Response

Dear Editors and Reviewers:

Thank you for your letter and comments relating to our manuscript entitled “Physicochemical Properties and Fungal Communities of Soil Types for Growth of the Medicinal Asian Water Plantain, Alisma orientale” (ID: jof- 2855766). The comments were very helpful in revising and improving our manuscript as well as emphasizing the significance to our research. We have read the comments carefully and made corrections accordingly. Revised portions are marked in blue in the manuscript. The main corrections in the paper and our responses to the reviewer’s comments are given below. We hope that the revisions in the manuscript and our accompanying responses will be sufficient to make our manuscript suitable for publication in the Journal of Fungi.

Responses to the comments of the reviewer:

Reviewer 2#

Major comments

Comments 1: Conducting this study, the authors did a lot of work on soil quality, fungal communities' composition, and the quality of medicinal plant Alisma orientale grown in the soil of four experimental plots. However, the significance of this work is greatly diminished by the low quality of results' presentation and their interpretation.

Response 1: We have improved the presentation and interpretation of results in the revised manuscript.

Comments 2: One of the major problem of the paper is its language, even not English by its own (although it needs to be thoroughly checked and corrected), but the scientific language. The way, in which the results of statistical analyses and the composition of fungal communities (all these "total fungal taxa") were described, arises great doubt about mycological background of the authors. 

Response 2: We have changed “total fungal taxa” to “OTUs” in the revision and have improved the revised manuscript for publication.

Detail comments

Comments 3: The problems start with the title. The title of the paper submitting to the mycological journal begins with "physicochemical properties", which is illogical. Then, here and further in the text – the authors did not measure any soil physical parameters (color, texture, structure, porosity, density, consistence, aggregate stability, and temperature), only chemical parameters. Also, here and further in the text – the term "soil type" is from the soil classification and is not appropriate in the context of the study. The title also should denote the region of the study. Overall, the expression "Physicochemical Properties and Fungal Communities of Soil Types for Growth …" is awkward.

Response 3: We would like to thank the reviewer’s valuable suggestion. We have changed the title in the revised manuscript.

Comments 4: Abstract is also problematic because of the following reason. Abstract should be understood by itself, without the full paper reading. However, this is not the case of the submitted abstract. The mention of soil fungal communities (line 21) arises suddenly, without any association with the previous text and the optimal condition for the plant growth. It is unclear what all these "soil groups" mean (lines 23-24). It is also unclear what Alismatis rhizome means.

Response 4: We have revised the Abstract in the revised manuscript.

Comments 5: Introduction should not contain all these unnecessary repetitions and general statements (see the attached PDF file).

Response 5: We have revised the Introduction.

Comments 6: Subsection 2.1 - natural vegetation and soil type (types) should be denoted.

Response 6: We have denoted the natural vegetation and soil types in 2.1.

Comments 7: Subsection 2.8. – statistical analysis is unclearly written and should be rewritten.

Response 7: We have rewritten statistical analysis in the revised version of the manuscript.

Comments 8: Subsection 3.1. - There is no need to repeat the written in M&M! It is more than enough to write that the chemical properties of studied soils are presented in Table 1.

Response 8: We have removed the repeated content in Subsection 3.1.

Comments 9: Subsection 3.3. - All these comparisons including "total fungal taxa" look meaningless, because "total taxa" is something borderless and vague.

Response 9: We have revised the comparisons in 3.3.

Comments 10: Figure 2. – The combination of four different figures in one makes each figure hardly readable. Figure A looks unclear and meaningless. Figure C just presents number of OTUs in the soil of studied plots (what does "feature number" mean?). Figure D is meaningful only if the numbers in the diagram ovals present numbers of OTUs.

Response 10: We revised Figure 2 in the revised manuscript.

Comments 11: Discussion suffers from the superficial interpretation of the results obtained, logical gaps, unnecessary repetitions, and problematic statements (see the attached PDF file).  It is unclear for which purposes the authors studied the composition of fungal communities at different taxonomic levels, if they did not discuss this aspect at all, except for the extremely general, problematic, and unclear statements on lines 485-489. It looks like that in order to choose the best plot for growing Alisma orientale, it was quite enough to conduct the experiment described in the subsection 3.8.

Response 11: We totally agree with your suggestions which might be of great help to improve the quality of our manuscript. According to your suggestions, we have revised the Discussion in the amended manuscript.

Comments 12: All other numerous comments, corrections, and suggestions are inserted into the attached PDF version of manuscript.

Response 12: We have revised all other numerous comments, corrections, and suggestions.

We tried our best to improve the manuscript and made some changes marked in blue in revised paper which will not influence the content and framework of the paper. We appreciate for Editors/Reviewers’ warm work earnestly and hope the revision will meet with your approval. Once again, thank you very much for your comments and suggestions.

Kind regards,

Junzhi Qiu

E-mail address: [email protected]

Reviewer 3 Report

Due to the choice of the MS topic, it has serious practical implications, since it can contribute to the development of the cultivation of a medicinal plant with a serious past and presumably still in demand in the future.

One of the basic ideas on which the thesis is based: "knowledge concerning for fungal community in different soils can lead to rational planting strategies for specific plants" (lines 84-86), is of particular importance, since the biomass of soil fungi even in the soils of agricultural areas is several dozen tons. This approach brought good results, for example, in renewing the cultivation of truffles in plantation.

The editing and wording of the MS (including chapter 2.1) is nice and thorough.

The following caused the reviewer a certain sense of lack. It is recommended to replace and present them with a single sentence or short data communication:

- Mycorrhizal status of Alisma orientale, literature related to its mycorrhizal formation

- ad table 1.: If a humus content measurement was made, it would be worthwhile to report it here. This parameter can have a very significant effect on some soil fungi (e.g. cultivated truffle species) and on the growth of plants.

- In the case of large fungi, the diversity of acidic soils is generally significantly higher, but the results in this MS do not reflect such a correlation (Table 1.& line 302.). It is recommended to briefly discuss this contradiction further, even by touching on the absorptive nutrition of fungal hyphae (to lines 490-504 ).

- The number of soil microbes can significantly decrease in the case of general or specific replant diseases. In the case of the test sites with lower diversity, did the possibility arise that the lower diversity was formed as a result of previous agricultural cultures, which could possibly be improved by treating soil fertility?

Additional suggestions for improvements:

Minor character errors, missing spaces: ad. 60, 69,

Write scientific names in italics: ad 579, 615, 646.

Author Response

Dear Editors and Reviewers:

Thank you for your letter and comments relating to our manuscript entitled “Physicochemical Properties and Fungal Communities of Soil Types for Growth of the Medicinal Asian Water Plantain, Alisma orientale” (ID: jof- 2855766). The comments were very helpful in revising and improving our manuscript as well as emphasizing the significance to our research. We have read the comments carefully and made corrections accordingly. Revised portions are marked in green in the manuscript. The main corrections in the paper and our responses to the reviewer’s comments are given below. We hope that the revisions in the manuscript and our accompanying responses will be sufficient to make our manuscript suitable for publication in the Journal of Fungi.

Responses to the comments of the reviewer:

Reviewer 3#

Detail comments

The following caused the reviewer a certain sense of lack. It is recommended to replace and present them with a single sentence or short data communication:

Comments 1: Mycorrhizal status of Alisma orientale, literature related to its mycorrhizal formation.

Response 1: We are deeply grateful for the reviewer to raise this important issue. As we know, it is disappointing to note the current lack of relevant research reports focusing on the mycorrhizal status of Alisma orientale.

Comments 2: ad table 1.: If a humus content measurement was made, it would be worthwhile to report it here. This parameter can have a very significant effect on some soil fungi (e.g. cultivated truffle species) and on the growth of plants.

Response 2: We totally agree with your suggestions which might be of great help to improve the quality of our manuscript. It is a pity that the humus content measurement for the soils was not made in this study. In the future, we would like to carry out more extensive experiment on this topic according to your suggestions.

Comments 3: In the case of large fungi, the diversity of acidic soils is generally significantly higher, but the results in this MS do not reflect such a correlation (Table 1.& line 302.). It is recommended to briefly discuss this contradiction further, even by touching on the absorptive nutrition of fungal hyphae (to lines 490-504 ).

Response 3: We have changed the corresponding part in the revised manuscript to make it clear (Discussion paragraph 2 line 490-498).

Comments 4: The number of soil microbes can significantly decrease in the case of general or specific replant diseases. In the case of the test sites with lower diversity, did the possibility arise that the lower diversity was formed as a result of previous agricultural cultures, which could possibly be improved by treating soil fertility?

Response 4: We highly appreciate the reviewer’s precious opinion. In this study, we examined the fungal diversity of four soils prior to the planting of Alisma orientale. The test sites with lower diversity, where only mulberry or tea trees have been cultivated (SST or CST), do not exhibit concerns regarding replanting diseases.

Additional suggestions for improvements:

Comments 5: Minor character errors, missing spaces: ad. 60, 69,

Response 5: we have added spaces in the corresponding part.

Comments 6: Write scientific names in italics: ad 579, 615, 646.

Response 6: We have revised scientific names in italics.

We tried our best to improve the manuscript and made some changes marked in blue in revised paper which will not influence the content and framework of the paper. We appreciate for Editors/Reviewers’ warm work earnestly and hope the revision will meet with your approval. Once again, thank you very much for your comments and suggestions.

Kind regards,

Junzhi Qiu

E-mail address: [email protected]

Round 2

Reviewer 1 Report

It is done all the ajustments necessary . 

It is done all the ajustments necessary .